# Influence of perceived threat of Covid-19 and HEXACO personality traits on toilet paper stockpiling

Lisa Garbe[1], Richard Rau[2], Theo Toppe[3] *

1 Institute for Political Science, School of Economics and Political Science, University of St.Gallen, St.Gallen, Switzerland, 2 Department of Psychology, University of Münster, Münster, Germany, 3 Department of Comparative Cultural Psychology, Max Planck Institute for Evolutionary Anthropology, Leipzig, Germany

☯ These authors contributed equally to this work.

* theo_toppe@eva.mpg.de

**Data Availability Statement:** All data files, supplementals, and R code are available from the Open Science Framework: https://osf.io/nbrg5/

**Funding:** The author(s) received no specific funding for this work.

## Abstract

Following the fast spread of Covid-19 across Europe and North America in March 2020, many people started stockpiling commodities like toilet paper. Despite the high relevance for public authorities to adequately address stockpiling behavior, empirical studies on the psychological underpinnings of toilet paper stockpiling are still scarce. In this study, we investigated the relation between personality traits, perceived threat of Covid-19, and stockpiling of toilet paper in an online survey (*N* = 996) across 22 countries. Results suggest that people who felt more threatened by Covid-19 stockpiled more toilet paper. Further, a predisposition towards Emotionality predicted the perceived threat of Covid-19 and affected stockpiling behavior indirectly. Finally, Conscientiousness was related to toilet paper stockpiling, such that individuals higher in Conscientiousness tended to stockpile more toilet paper. These results emphasize the importance of clear communication by public authorities acknowledging anxiety and, at the same time, transmitting a sense of control.

## Introduction

Within a few weeks, the outbreak of the Covid-19 pandemic has turned into a severe global health crisis in spring 2019 [1]. With the increasing spread of the virus, the demand for particular commodities such as toilet paper has skyrocketed. Some companies reported an increase of up to 700% in their sales [2–4]. Despite government appeals to refrain from "panic buying" or stockpiling [5], supermarkets across countries face difficulties in stocking up toilet paper. The resulting scarcity of toilet paper in some households has led to problematic consequences such as the clogging of outfall pipes after people started using products other than toilet paper [6]. In response to the increased stockpiling of toilet paper across countries, numerous media articles sought to explain its underlying psychological processes [7–12]. However, to date, most claims are hardly supported by empirical evidence despite recent calls for more social and behavioral studies to support effective strategies in response to Covid-19 [13]. In this

**Competing interests:** The authors have declared that no competing interests exist.

study, we examined the relationship between personality traits, perceived threat of Covid-19, and the hoarding of toilet paper to learn more about its psychological underpinnings.

Which individual difference variables can account for toilet paper hoarding? On a superficial level, private stockpiling of limited resources may appear first and foremost as an instance of selfishness. In fact, there exist stable and substantial individual differences in peoples' concern for their own vs. everyone's welfare [14] and these differences are frequently found to explain prosocial vs. antisocial behavior in contexts involving a shortage of resources [15]. Another explanation that has been prominently featured in the media revolves around an overgeneralization of disgust. According to this notion, people experience an increased sensitivity to disgust in times of a spreading disease [10] and toilet paper is hypothesized to serve as a symbol of safety alleviating the perceived threat [11]. Consequently, stockpiling toilet paper during the Covid-19 pandemic should be observed primarily among those who feel particularly threatened by the virus. Although stockpiling as a result of perceived threat might be considered selfish by some, it is important to note that it would not necessarily reflect a dispositional lack of prosociality. Instead, even the most humble and moral individuals might stockpile toilet paper as long as they feel sufficiently threatened by the pandemic. Finally, stockpiling toilet paper has also been interpreted in terms of classic psychoanalytic theory. In this line of reasoning, individuals with a marked pattern of orderliness and self-discipline, or an "anal-retentive personality" [16], are hypothesized to be particularly inclined to hoard toilet paper [8,17]. At the same time, however, these individuals may also exhibit high levels of self-control and may refrain from impulsive panic-purchases more easily.

In order to examine individual differences underlying toilet paper consumption empirically, we attended to the HEXACO model of personality in the present study. The HEXACO model is rooted in lexical studies of personality descriptors across various languages and organizes individual differences along six broad personality domains [18]: Honesty-Humility (characterized by the facets sincerity, fairness, greed avoidance, and modesty), Emotionality (fearfulness, anxiety, dependence, sentimentality), eXtraversion (social self-esteem, social boldness, sociability, liveliness), Agreeableness (forgiveness, gentleness, flexibility, patience), Conscientiousness (organization, diligence, perfectionism, prudence), and Openness to Experience (aesthetic appreciation, inquisitiveness, creativity, unconventionality). As a widely established and well-validated taxonomy, the HEXACO model allowed us to empirically address many of the speculations expressed in the popular media during the Covid-19 outbreak in an objective and methodologically sound manner. Differences in people's solidary concern for others' is captured in the Honesty-Humility dimension, differential tendencies to worry and be anxious are captured in the Emotionality dimension, and differences in orderliness and self-discipline are captured in the Conscientiousness dimension. Although we did not have specific reasons to expect relations between toilet paper consumption and the remaining HEXACO dimensions, we considered these as well for completeness.

To the best of our knowledge, only one study has so far examined the relationship between personality traits and hoarding behavior during the Covid-19 pandemic. This study focused exclusively on the Honesty-Humility dimension and found initial evidence that hoarding was driven by a lack of solidarity in a sample of UK residents [19]. However, no empirical study exists on the link between toilet paper stockpiling and the remaining personality domains which leaves the role of personality, defined more broadly, unanswered. In this study, we surveyed an international sample of adults to explore the relationships between the HEXACO personality dimensions, experiences of threat of Covid-19, and toilet paper consumption.

A more comprehensive understanding of how the perceived threat of Covid-19 and different personality traits trigger stockpiling behavior has important implications for public policies directed at households and individuals as well as grocery stores selling toilet paper and

other scarce commodities. To date, governments and companies have implemented different measures to guarantee comprehensive supply and a deeper psychological understanding of toilet paper stockpiling will help to evaluate and improve these measures.

## Methods

Our study was conducted from March 23rd to March 29th, 2020, a period in which the total number of confirmed cases of Covid-19 increased from about 378.200 to more than 650.000 [20]. Many national governments implemented partial or complete lockdowns during that time (e.g., Germany or United States). Thus, this period was characterized by frequent and drastic changes in public life and was accompanied by immediate shortages of resources such as toilet paper. Due to the exploratory nature of our research question, our sampling strategy was not based on power considerations to detect a specific effect. Instead, we aimed for a sample size of $N = 1,000$ as this would yield sufficient power (90%) to detect small effects ($r = .10$) in a two-sided test at an alpha-rate of 5%.

### Participants

In total, $N = 1,029$ adults from 35 countries took part in the study. The survey was advertised via mailing lists and postings on social media platforms. Participation was anonymous and voluntary and participants did not receive any incentives. Before taking the survey, participants provided written informed consent by confirming that their participation was voluntary, that they understood the study's goals, and that they knew that they could withdraw from participation at any time. We classified participants with respect to their place of residence (Europe, United States/Canada, Other). Participants in the "Other" category were excluded from the analyses due to the small size of that category (see S1 Table in the Supporting Information for the sample size of each country). Table 1 provides a description of the final sample ($N = 996$ participants from 22 countries).

### Materials and procedure

After providing informed consent, participants filled out the Brief HEXACO Inventory (BHI; [21]) which comprises four items for each of the six dimensions (one item per facet, 24 items in total). They then indicated their currently perceived level of threat posed by Covid-19 on a 10-point visual analogue scale and provided information about current curfew regulations at their place of residence (e.g., availability of local public transport; see Table 1). Further, participants described their toilet paper (ToP) consumption behavior. In particular, participants indicated (a) their ToP shopping frequency in the past two weeks (*not once, once, twice, three times or more*), (b) their ToP shopping intensity, i.e. how many packages of toilet rolls they bought (*none, one, two, three or more*), (c) the amount of toilet rolls currently stocked in their household (*none; 1 to 4; 5 to 8; 9 to 12; 13 to 16; 17 to 20; 21 or more*), and (d) whether they had stocked an unusual amount of toilet paper (*less than usual, usual, more than usual*). Then, participants indicated whether and for how long their household had been in strict quarantine (i.e., not leaving the house at all) as well as how many high-risk persons live in their household (e.g., due to age or pre-existing condition). Finally, participants reported their age, gender, place of residence, nationality, household size, as well as their political left-right placement on a 11-point visual analog scale [22]. We created an English and a German version of this questionnaire (see S2 and S3 Tables for both versions of the questionnaire). All item translations were discussed with native speakers of both languages and any disagreements were resolved. The questionnaire was implemented via the online survey platform *formr* [23].

**Table 1. Descriptive statistics by place of residence.**

| Variable | Measure | Value | |
|---|---|---|---|
| | | US/Canada (n = 267) | Europe (n = 729) |
| | M (SD) | | |
| Age in years | | 32.39 (10.64) | 32.09 (9.36) |
| Gender | % | | |
| Female | | 81.64 | 68.31 |
| Male | | 15.73 | 30.59 |
| Diverse | | 2.62 | 1.10 |
| | M (SD) | | |
| Household size | | 2.55 (1.18) | 2.61 (1.70) |
| Days in quarantine | | 2.30 (4.05) | 0.99 (3.17) |
| High risk people in household[*] | | 0.41 (0.76) | 0.28 (0.65) |
| Days between participation and first recorded case of Covid-19 in country | | 63.08 (1.67) | 54.80 (9.70) |
| Political left right placement | | 3.00 (1.86) | 3.33 (1.59) |
| | % yes | | |
| Personal mobility restriction | | 60.67 | 84.64 |
| Leaving the house is only permitted in specific professions[*] | | 95.68 | 76.82 |
| Leaving the house is only permitted in small groups[*] | | 69.75 | 95.14 |
| Restrictions on public life[*] | | 98.13 | 99.04 |
| Educational facilities are closed[*] | | 100 | 99.86 |
| Restaurant, bars, cafés are closed[*] | | 93.89 | 99.31 |
| Local public transport is restricted | | 52.43 | 65.84 |
| | M (SD) | | |
| Toilet paper packages bought | | 1.85 (0.92) | 1.81 (0.70) |
| Shopping frequency | | 1.63 (0.63) | 1.71 (0.59) |
| Toilet rolls in household | | 12.47 (6.11) | 8.90 (5.27) |
| | % | | |
| Current amount of toilet paper is. . .[*] | | | |
| Less than usual | | 9.36 | 9.74 |
| Usual | | 73.41 | 76.54 |
| More than usual | | 17.23 | 13.71 |
| | M (SD) | | |
| Perceived threat by Covid-19 | | 6.05 (1.99) | 4.59 (2.15) |
| HEXACO Dimensions | | | |
| Honesty-Humility | | 4.06 (0.57) | 3.76 (0.68) |
| Emotionality | | 3.08 (0.66) | 3.01 (0.64) |
| Extraversion | | 3.81 (0.62) | 3.97 (0.64) |
| Agreeableness | | 2.92 (0.62) | 2.85 (0.56) |
| Conscientiousness | | 3.60 (0.66) | 3.42 (0.61) |
| Openness | | 3.92 (0.59) | 4.02 (0.52) |

*M*, mean; *SD*, standard deviation

[*]Excluded from analysis due to limited variance.

## Data analytic approach

Initial inspection of the data revealed that some variables had limited variance and, thus, were of limited explanatory value for our analyses (see Table 1). These variables pertained to curfew regulations at the participants residence (e.g., whether schools or restaurants are closed), to the

proportion of high-risk persons in the participants' households, and to the (un)usualness of the currently stocked amount of toilet paper and are not considered in the remainder of the article. ToP shopping frequency and ToP shopping intensity were strongly correlated with one another ($r = .80$) but only weakly correlated with the amount of stocked toilet rolls ($r$s = .27 and .21, respectively; see S4 Table for correlations of all variables included in the statistical models). Given this heterogeneity, we examined our research question separately for each of these ToP consumption indicators.

We analyzed the data in a series of multiple regressions aimed at explaining the perceived threat of Covid-19 first, and stockpiling of toilet paper second. For each dependent variable, we first computed a baseline model featuring several control variables that could presumably be related to that variable without being of psychological interest in their own right. These control variables were age, gender (female vs. male vs. diverse), household size, personal mobility restrictions (yes vs. no), restrictions on public transport (yes vs. no), number of days in strict quarantine, political left-right placement, residence (US/Canada vs. Europe), and the number of days between participation and the first recorded case of Covid-19 in the participants' residence (retrieved from http://www.worldometers.info/). We then entered the psychological predictor of interest in a second step to examine its effect above and beyond the control variables of the baseline model. As the psychological predictors, we considered the six HEXACO dimensions and, when predicting ToP consumption, the perceived threat of Covid-19. Thus, each baseline model was followed by six (when perceived threat was the dependent variable) or seven (when a ToP consumption indicator was the dependent variable) models each of which addressed the unique predictive value of one psychological variable.

Whenever we found a significant effect in this step, we allowed for an interaction with residence to test whether the effect was moderated by participants' place of residence as a follow-up analysis. To ease interpretation, continuous variables were z-standardized and categorical variables were dummy-coded in all models. All analyses were conducted in R [24]. Data and R-code are available at https://osf.io/nbrg5/.

## Results

### Measurement characteristics of the BHI

To make sure that our German translation of the BHI captured the same latent constructs as the original English version, we tested measurement invariance for each personality dimension. Specifically, the fact that we were interested in the relation between personality and our dependent variables (but not in country-level differences in personality) required metric (but not scalar) equivalence. Thus, we compared a metric model in which indicators were constrained to be equal across versions with a configural model in which the four indicators of a dimension were estimated freely across versions. The metric model was supported for each of the six HEXACO dimensions, $\Delta\chi^2(3) < 6.58$, $p > .08$ for all model comparisons. Internal consistencies were modest ($\alpha_{HH} = .50$; $\alpha_E = .37$; $\alpha_X = .64$; $\alpha_A = .36$; $\alpha_C = .54$; $\alpha_O = .49$) as is expected in short instruments that seek to maximize content validity [21].

### Perceived threat of Covid-19

The baseline model for perceived threat of Covid-19 revealed that the likelihood to feel threatened increases significantly with age ($p = .019$; see Table 2) and with the number of days spent in quarantine ($p = .002$). Female participants felt more threatened by Covid-19 than male participants ($p = .001$). Moreover, participants residing in Europe reported to feel significantly less threatened than their North-American counterparts ($p < .001$). The models for the

**Table 2. Prediction of perceived threat of Covid-19 and toilet paper stockpiling.**

| Predictors | Dependent Variable | | | | | | | |
|---|---|---|---|---|---|---|---|---|
| | Perceived Threat of Covid-19 | | ToP Shopping Frequency | | ToP Shopping Intensity | | Stocked ToP | |
| Baseline Model | b | SE | b | SE | b | SE | b | SE |
| Age | **0.072** | **0.030** | **0.085** | **0.032** | **0.099** | **0.032** | **0.069** | **0.031** |
| Female gender (ref: male) | **-0.226** | **0.069** | -0.028 | 0.073 | 0.016 | 0.073 | -0.063 | 0.070 |
| Other gender (ref: male) | -0.355 | 0.248 | -0.224 | 0.262 | -0.102 | 0.261 | 0.256 | 0.249 |
| Household size | 0.012 | 0.030 | 0.053 | 0.032 | **0.069** | **0.032** | 0.044 | 0.030 |
| Personal mobility restriction | -0.125 | 0.075 | 0.055 | 0.080 | 0.080 | 0.080 | -0.087 | 0.076 |
| Public transport restriction | -0.021 | 0.063 | -0.065 | 0.066 | -0.080 | 0.080 | -0.065 | 0.063 |
| Days in quarantine | **0.097** | **0.031** | 0.061 | 0.032 | 0.049 | 0.032 | 0.031 | 0.031 |
| Political orientation (left to right) | -0.007 | 0.031 | -0.008 | 0.032 | 0.013 | 0.032 | **0.127** | **0.031** |
| Place of residence | **-0.614** | **0.079** | **0.172** | **0.083** | -0.055 | 0.083 | **-0.642** | **0.080** |
| Days since first Covid-19 case | -0.006 | 0.004 | 0.000 | 0.004 | -0.001 | 0.004 | -0.055 | 0.033 |
| Incremental main effects of psychological variables | b | SE | b | SE | b | SE | b | SE |
| Perceived threat of Covid-19 | — | — | **0.076** | **0.033** | **0.077** | **0.033** | **0.100** | **0.032** |
| Honesty-Humility | 0.026 | 0.032 | -0.002 | 0.033 | -0.008 | 0.033 | 0.045 | 0.032 |
| Emotionality | **0.188** | **0.031** | 0.041 | 0.033 | 0.018 | 0.033 | 0.039 | 0.032 |
| Extraversion | -0.039 | 0.030 | 0.012 | 0.032 | 0.018 | 0.032 | -0.009 | 0.031 |
| Agreeableness | -0.026 | 0.030 | -0.020 | 0.032 | -0.004 | 0.032 | 0.001 | 0.031 |
| Conscientiousness | -0.047 | 0.030 | 0.059 | 0.032 | **0.064** | **0.032** | **0.061** | **0.031** |
| Openness to experience | -0.039 | 0.030 | 0.000 | 0.032 | 0.040 | 0.032 | **-0.075** | **0.030** |
| Interaction effects of psychological variables and place of residence | b | SE | b | SE | b | SE | b | SE |
| Perceived threat of Covid-19 | | | 0.136 | 0.078 | 0.091 | 0.078 | 0.071 | 0.074 |
| Honesty-Humility | | | | | | | | |
| Emotionality | 0.101 | 0.066 | | | | | | |
| Extraversion | | | | | | | | |
| Agreeableness | | | | | | | | |
| Conscientiousness | | | 0.087 | 0.070 | 0.112 | 0.070 | 0.004 | 0.066 |
| Openness to experience | | | | | | | 0.010 | 0.065 |
| Upper limit model determination[a] | $R^2 = .147$ | | $R^2 = .024$ | | $R^2 = .024$ | | $R^2 = .116$ | |

Significant regression weights (p < .05) are printed in bold. Interaction effects were only tested for models that involved a significant main effect for the psychological variable at hand. ToP = Toilet Paper. Coding of place of residence: 0 = US/Canada; 1 = EU.

[a] Model determination is presented for the model with the most influential psychological variable for the respective dependent variable.

HEXACO dimensions revealed a link between Emotionality and the perceived threat of Covid-19 ($p < .001$) with participants higher on Emotionality reporting more perceived threat.

## Toilet paper consumption

For all variables indicating toilet paper consumption—shopping frequency, shopping intensity, and number of stocked toilet rolls—the baseline models revealed a positive relation with age ($ps < .009$; see Table 2). That is, older participants shopped more frequently, bought more packages of toilet paper and had more toilet papers rolls in stock as compared to younger participants. Participants residing in Europe shopped toilet paper more frequently than North-American residents ($p = .039$) but had less toilet paper in stock ($p < .001$).

Turning to the psychological predictors, the perceived threat of Covid-19 was positively related to all three ToP variables ($p$s < .025). Participants who reported to feel more threatened shopped toilet paper more frequently, bought more packages, and had more toilet paper in stock. Also, the models suggested that Conscientiousness is positively associated with toilet paper consumption. In particular, participants high on Conscientiousness tended to shop more frequently ($p$ = .065), shopped more ToP ($p$ = .045), and stocked more toilet paper ($p$ = .048).

Following up on these findings by allowing for an interaction term of the respective predictor with participants' place of residence did not yield any significant effects (all $p$s > .08). That is, associations between psychological variables on the one hand and perceived threat of Covid-19 and ToP consumption on the other hand did not differ systematically for USA/Canada vs. EU residents.

## Indirect effects of emotionality on toilet paper consumption

Given the link between Emotionality and perceived threat posed by Covid-19 on the one hand, and perceived threat and toilet paper consumption on the other, we examined the indirect effect of Emotionality on toilet paper consumption through threat perception. We therefore re-estimated the models that involved Emotionality and perceived threat as predictors with the lavaan package [25] and used bootstrapped confidence intervals to evaluate the significance of the indirect effect.

For toilet paper shopping intensity and the amount of stocked toilet rolls, we found that the indirect effect of Emotionality through perceived threat was significant (c' = .016; 95% CI = [0.002; 0.031] for ToP shopping intensity; c' = .019; 95% CI = [-0.006; 0.036] for stocked toilet rolls; see Fig 1). The indirect effects for ToP shopping frequency was marginally significant (c' = .014; 95% CI = [0.001; 0.029]; see Fig 1). These results suggest that Emotionality may fuel the feeling of being threatened by the Covid-19 pandemic which may consequently foster toilet paper stockpiling.

## Discussion

The three main findings of the current study are the following: First, the perceived threat of Covid-19 predicts toilet paper stockpiling. Second, Emotionality predicts the perceived threat of Covid-19 and thereby indirectly affects stockpiling behavior. Third, individuals high in Conscientiousness engage in more toilet paper stockpiling. All these effects held across North American and European countries and were robust across different indicators of toilet paper stockpiling (i.e., shopping frequency, shopping intensity, and stocked toilet rolls). Importantly, we ruled out that these effects were driven by socio-demographic characteristics (i.e., age, gender, household size, political attitudes) or by regulations of local authorities (i.e., restrictions of personal mobility or public transport).

The most robust predictor of toilet paper stockpiling was the perceived threat posed by Covid-19. People who feel more threatened by the pandemic stockpile more toilet paper. Given that stockpiling is objectively unrelated to saving lives or jobs during a health crisis, this finding supports the notion that toilet paper functions as a purely subjective symbol of safety.

We also found that this effect was partly based on the personality factor of Emotionality. Around 20 percent of the differences in toilet paper consumption that were explained by feelings of threat were based on people's dispositional tendency to worry a lot and generally feel anxious. At the same time, the remaining 80 percent of this effect were not found to be rooted in personality differences. This suggests that how much people feel personally threatened by Covid-19 also depends on psychological factors not accounted for in our study or on malleable

a

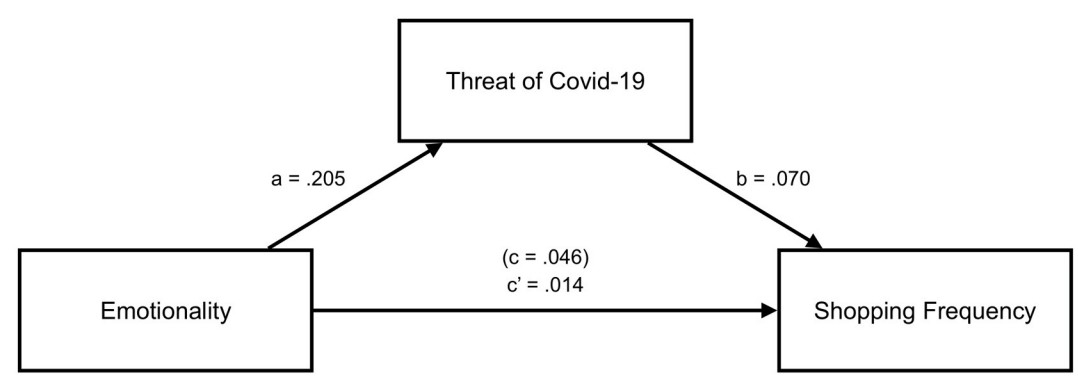

b

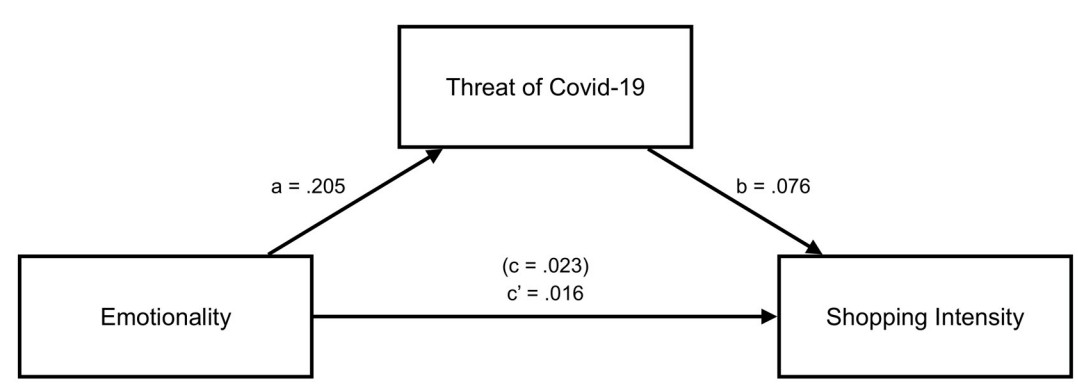

c

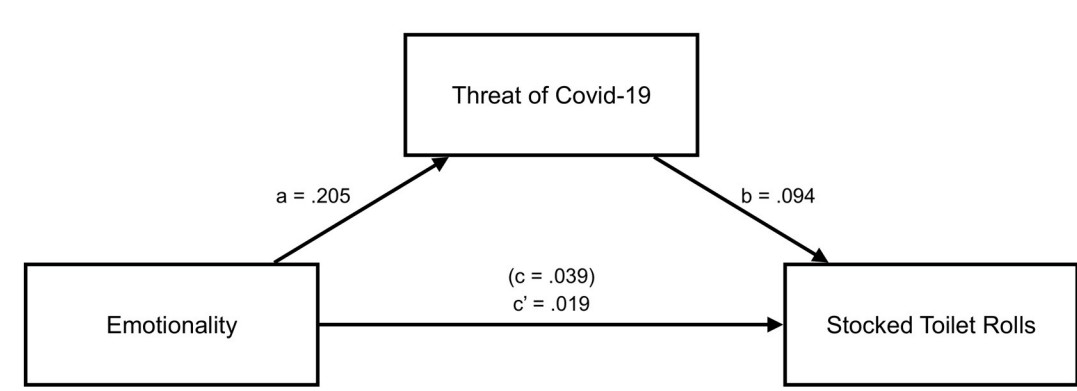

**Fig 1.** Panels indicate the indirect effects of Emotionality on (a) shopping frequency, (b) shopping intensity, and (c) stocked toilet paper rolls. In each panel, c refers to the total effect of Emotionality and c' refers so the indirect effect of Emotionality through the perceived threat of Covid-19.

external factors such as the risk management by and trust in local authorities. Hence, these findings highlight the potential of public communication to address individuals' perceptions of threat and thereby alter their shopping behavior. For instance, research on communication strategies suggests that clear communication aiming to increase awareness of a disease and providing simple behavioral instructions reduces people's threat perception [26]. While it is important to communicate the severity of a pandemic and appeal to people's compliance to necessary measures such as social distancing, communicators should be careful not to provoke panic that can eventually result in dysfunctional behavior such as stockpiling (see also [27]). This is also in line with the finding that fear can potentially be useful if people "feel capable of dealing with the threat" ([13], p. 2). If fear is driven by strong emotions, however, people may ignore factual information and engage in irrational behavior (ibid.).

In addition to the effect of perceived threat, we found personality differences in Conscientiousness to be another robust predictor of toilet paper stockpiling. More conscientious people tend to stockpile more toilet paper. This finding is in line with the expectation that long-sighted and more orderly individuals engage in more stockpiling and does not support the counternarrative that conscientious individuals refrain from impulsive panic buying due to increased self-control. This finding implies that public communication is well advised to stress the functioning of supply chains and the long-term availability of vital commodities. Such rational appeal might exploit people's long-sightedness and effectively counter the dysfunctional intuition that commodities may become scarce in the near future.

In contrast to preliminary insights from Columbus' study [19], we did not find Honesty-Humility to be a significant predictor of toilet paper stockpiling. This implies that toilet paper stockpiling might not be resulting from a lack of solidarity and, as such, moral appeals by public authorities asking people to refrain from stockpiling might be less fruitful than expected. However, the discrepancy between the present findings and the ones from Columbus' study [19] call for further scrutiny given that there were major differences between the two studies from which diverging results may have emerged. For instance, we focused on toilet paper, surveyed participants from 22 countries, and controlled for several socio-demographic and local regulatory differences. In contrast, Columbus examined hoarding behavior more broadly, focused on a UK-only sample, and did not control for third variables. More research is needed to reveal which of these differences can resolve the discrepant findings with respect to the role of Honesty-Humility and stockpiling.

Our analyses further revealed that with increasing age, people tend to stockpile more toilet paper. Older people are more prone to a severe course of the disease and, thus, may be more eager to prepare for strict self-isolation. In addition, in some countries, older people were asked to self-isolate before more comprehensive lockdowns were put in place (e.g., [28]) which might partly account for the age effect. Besides, our results revealed differences between American and European households. As compared to European participants, Americans reported a higher perception of threat of Covid-19 which might result from different communication strategies of public authorities or differences in public health systems. Also, Americans stockpiled more toilet paper in their household and went toilet paper shopping less frequently as compared to Europeans. This could be attributed to the circumstance that, on average, toilet paper rolls come in bigger packages in the US (e.g., up to 36 rolls per package) than in most European countries (e.g., between 8 to 16 rolls).

While the current study provides valuable first insights into the psychological underpinnings of toilet paper stockpiling in the wake of a health crisis, some limitations will need to be addressed in future research. For example, future studies might examine regional or even local differences in personality effects on stockpiling by systematically sampling participants from areas that are differentially affected by the pandemic and/or from urban vs. rural areas. Although we did not observe differential psychological effects when comparing Europe vs. North America, it is conceivable that regional differences with respect to the severity of the health problems and to governmental responses will afford different psychological adaptations. Data with higher power and a higher spatial resolution will be needed to unravel such moderation effects. Second, future studies might consider individual differences outside of the HEX-ACO framework. Here, the considered variables explained only up to 12% of the variation in toilet paper consumption which suggests that some psychological explanations of toilet paper consumption have likely remained unaccounted for. Future studies might thus consider more narrow traits that are perhaps more immediately involved in motivating toilet paper stockpiling (or a lack thereof) such as optimism [29] or perfectionism [30] or that directly tap into antisocial tendencies such as the dark triad traits [31]. Also, experimental studies would be required in order to explicitly test the directionality implied in our investigation of indirect effects. Finally, a more detailed analysis of situational factors such as the increase of Covid-19 cases on a particular day or the communication strategies of local authorities might be promising avenues for explaining toilet paper stockpiling more comprehensively. Meanwhile, the present study suggests that low anxiety and little desire to plan ahead are the best psychological protective factors to refrain from irrationally stockpiling limited resources in times of a health crisis.

## Supporting information

**S1 Table. Sample sizes for all residences.**
(DOCX)

**S2 Table. English version of the questionnaire.**
(DOCX)

**S3 Table. German version of the questionnaire.**
(DOCX)

**S4 Table. Correlations with confidence intervals.**
(DOCX)

## Acknowledgments

We thank all employees in supermarkets who are working under enormous pressure at the moment for their valuable work. We thank Sabine Ertel for her inspiration to investigate toilet paper hoarding and Florian Scharf for patiently responding to voice messages. Also, we thank Valerio Capraro and an anonymous reviewer for fruitful comments and a rapid review process. Finally, we thank all participants.

## Author Contributions

**Conceptualization:** Lisa Garbe, Richard Rau, Theo Toppe.

**Data curation:** Lisa Garbe, Richard Rau, Theo Toppe.

**Formal analysis:** Lisa Garbe, Richard Rau, Theo Toppe.

**Investigation:** Lisa Garbe, Richard Rau, Theo Toppe.

**Methodology:** Lisa Garbe, Richard Rau, Theo Toppe.

**Visualization:** Lisa Garbe, Richard Rau, Theo Toppe.

**Writing – original draft:** Lisa Garbe, Richard Rau, Theo Toppe.

**Writing – review & editing:** Lisa Garbe, Richard Rau, Theo Toppe.

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
