## [Decision Letter · Decision Letter 0]

4 May 2020

PONE-D-20-10675

Influence of perceived threat of Covid-19 and HEXACO personality traits on toilet paper stockpiling

PLOS ONE

Dear Mr. Toppe,

Thank you for submitting your manuscript to PLOS ONE. After careful consideration, we feel that it has merit but does not fully meet PLOS ONE’s publication criteria as it currently stands. Therefore, we invite you to submit a revised version of the manuscript that addresses the points raised during the review process.

We would appreciate receiving your revised manuscript by Jun 18 2020 11:59PM. To enhance the reproducibility of your results, we recommend that if applicable you deposit your laboratory protocols in protocols.io, where a protocol can be assigned its own identifier (DOI) such that it can be cited independently in the future. For instructions see: http://journals.plos.org/plosone/s/submission-guidelines#loc-laboratory-protocols

We look forward to receiving your revised manuscript.

Kind regards,

Valerio Capraro

Academic Editor

PLOS ONE

Additional Editor Comments (if provided):

I have collected one review from one expert in the field. The reviewer likes the paper and suggests minor revision. I've read the paper myself and I agree with the opinion of the reviewer. Therefore, I would like to invite you to revise your work following the reviewer's comments. Additionally, I would like to note that, just days ago, Van Bavel et al. published on Nature Human Behaviour a "perspective article" on what social and behavioural science can do to promote pandemic response. I think this could be a useful reference, given the relevance for your work.

Van Bavel, et al. (2020). Using social and behavioural science to support COVID-19 pandemic response. Nature Human Behaviour.

2. Please provide additional details regarding participant consent. In the Methods section, please ensure that you have specified (1) whether consent was informed and (2) what type you obtained (for instance, written or verbal). If your study included minors, state whether you obtained consent from parents or guardians.

3. Please include additional information regarding the survey or questionnaire used in the study and ensure that you have provided sufficient details that others could replicate the analyses. For instance, if you developed a questionnaire as part of this study and it is not under a copyright license more restrictive than CC-BY, please include a copy, in both the original language and English, as Supporting Information

Reviewers' comments:

Reviewer's Responses to Questions

**Comments to the Author**

1. Is the manuscript technically sound, and do the data support the conclusions?

Reviewer #1: Yes

2. Has the statistical analysis been performed appropriately and rigorously? 

Reviewer #1: No

3. Have the authors made all data underlying the findings in their manuscript fully available?

Reviewer #1: Yes

4. Is the manuscript presented in an intelligible fashion and written in standard English?

Reviewer #1: Yes

5. Review Comments to the Author

Reviewer #1: This study is pretty unique and relevant to understand hoarding behaviour during the COVID-19 pandemic. The manuscript reads well and the mechanisms suggested by the authors are well explained. There are some explanations that are worth including as the readers could learn more of the rich data the authors have collected. I hope the authors continue expanding this line of research in the near future.

Data collection:

1. Authors are clear about the methods for data collection.

2. Authors are clear with how they collected their data.

3. Add in the sample description personality traits and threat variable. The authors have quite a rich data and the reader would be benefited from having summary statistics by countries if the number of observations allows it. If this is not possible, I suggest the authors to provide a comparison between Canada/USA vs Europe.

Data analysis:

1. Good practice to test measurement invariance for each personality dimension for both German and English versions.

2. When modelling, the authors need to consider the differences across countries based on the exposure to the disease (distance between data collection and the first case reported in the country). This new variable can also facilitate the authors to explore to what extent “perceived threat” is correlated with the exposure of the disease in the country of residence. Lockdown is correlated with this variable, but it is not necessary controlling for the level of exposure to the disease, those countries that got exposed much later had the chance to learn from the ones initially exposed. This can also explain why Europe shopped more than North American residents as the level of uncertainty was higher.

3. Add models in the manuscript as this will ease the understanding of your results. In particular, make explicit in your models the reference categories when using interactions. When analysing interactions, explain the results based on your reference categories.

4. The relationship between threat/emotionality and toilet paper shopping is difficult to understand in Figure 1, see the dispersion of the points. What do they mean? Can the authors offer a different way to present these results? Are these bivariate regressions?

5. The authors need to consider per capita stock of toilet paper, instead of stock, as this considers the natural demand for toilet paper given the household size. In the manuscript, it is not clear whether the authors considers a per capita measure or not.

6. In 263-264 lines, do you mean participants who are more open to experiences stocked less toilet rolls than those who present lower level of openness? If so, rewrite this lines to make this point clearer.

Further suggestions:

1. Can the authors provide an official reference for the content in line 41?

2. Despite the little variation of variables explained in lines 143-148, I recommend the authors to explain a bit more on which values/categories the data was concentrated the most.

3. Authors are finding associations, not impact or effects. The reviewer strongly recommends adjusting the language reflecting this (for instance, see lines 195 and 198).

4. Could the authors clarify whether those people who tend to stockpile as a result of the uncertainty of the consequences and causes of the disease can be considered selfish? If not, where would this explanation enter the reasons outlined in pg. 3.

5. Could the authors offer a brief discussion of what sort of framing governments could use when delivering messages to deter stockpiling during crisis?

6. PLOS authors have the option to publish the peer review history of their article (what does this mean?). If published, this will include your full peer review and any attached files.

Reviewer #1: No

---

## [Author Response · Author response to Decision Letter 0]

19 May 2020

Editor

Comment 1: “(…) I would like to note that, just days ago, Van Bavel et al. published on Nature Human Behaviour a "perspective article" on what social and behavioural science can do to promote pandemic response. I think this could be a useful reference, given the relevance for your work.”

Response: We thank you for pointing us to this important publication. In the revised manuscript we refer to the article both in the Introduction as well as in the Discussion:

• “However, to date, most claims are hardly supported by empirical evidence despite recent calls for more social and behavioral studies to support effective strategies in response to Covid-19 [13].” (p. 3)

• While it is important to communicate the severity of a pandemic and appeal to people’s compliance to necessary measures such as social distancing, communicators should be careful not to provoke panic that can eventually result in dysfunctional behavior such as stockpiling (see also [27]). “This is also in line with the finding that fear can potentially be useful if people “feel capable of dealing with the threat” ([13], p. 2). If fear is driven by strong emotions, however, people may ignore factual information and engage in irrational behavior (ibid.).” (p. 17)

Comment 2: “Please provide additional details regarding participant consent. In the Methods section, please ensure that you have specified (1) whether consent was informed and (2) what type you obtained (for instance, written or verbal). If your study included minors, state whether you obtained consent from parents or guardians.”

Response: Thank you for pointing out this unclarity. We revised the Participants section and gave more information about the consent (see p. 6). The Participants section now reads: “In total, N = 1,029 adults from 35 countries took part in the study. The survey was advertised via mailing lists and postings on social media platforms. Participation was anonymous and voluntary and participants did not receive any incentives. Before taking the survey, participants provided written informed consent by confirming that their participation was voluntary, that they understood the study’s goals, and that they knew that they could withdraw from participation at any time.”

Comment 3: “Please include additional information regarding the survey or questionnaire used in the study and ensure that you have provided sufficient details that others could replicate the analyses. For instance, if you developed a questionnaire as part of this study and it is not under a copyright license more restrictive than CC-BY, please include a copy, in both the original language and English, as Supporting Information”

Response: We think this is a good suggestion and have added the English and German version of the questionnaire to the Supporting Information (see Table S1 and S2). 

 

Reviewer 1

Comment 1: “Add in the sample description personality traits and threat variable. The authors have quite a rich data and the reader would be benefited from having summary statistics by countries if the number of observations allows it. If this is not possible, I suggest the authors to provide a comparison between Canada/USA vs Europe.”

Response: We agree that providing more details on summary statistics in a prominent location of the manuscript adds a valuable contribution. We have thus added a table (Table 1, p. 6-7) with descriptive statistics for all assessed variables separately for Europe and North-America. 

Comment 2: “When modelling, the authors need to consider the differences across countries based on the exposure to the disease (distance between data collection and the first case reported in the country). This new variable can also facilitate the authors to explore to what extent “perceived threat” is correlated with the exposure of the disease in the country of residence. Lockdown is correlated with this variable, but it is not necessary controlling for the level of exposure to the disease, those countries that got exposed much later had the chance to learn from the ones initially exposed. This can also explain why Europe shopped more than North American residents as the level of uncertainty was higher.”

Response: We thank the reviewer for this interesting comment. In the revised manuscript, we have included a variable indicating the number of days between the first reported case of Covid-19 in the participant’s country and the date of participation. However, the inclusion of this variable did not substantially change the pattern of results (see Results section; p. 10-13). 

Comment 3: “Add models in the manuscript as this will ease the understanding of your results. In particular, make explicit in your models the reference categories when using interactions. When analyzing interactions, explain the results based on your reference categories.”

Response: We agree that a more comprehensive presentation of the regression models is a good way to better communicate the findings We thus added a table (Table 2, see p. 12) providing the crucial estimates of all multiple regressions including an indication of reference categories. We believe that presenting all relevant results in one spot is a big asset and makes it much easier for readers to quickly get a sense of the pattern of results. When adding the number of days between the first recorded Covid-19 case and participation, there was no significant interaction between the psychological variables and the participants’ residence (US/Canada vs. Europe; see p. 13). Therefore, we do not describe the results of these non-significant interaction effects thoroughly. 

Comment 4: “The relationship between threat/emotionality and toilet paper shopping is difficult to understand in Figure 1, see the dispersion of the points. What do they mean? Can the authors offer a different way to present these results? Are these bivariate regressions?”

Response: The figure did in fact show the predicted values from a bivariate regression but we fully agree that the visualization was potentially misleading. Given our choice to include a Table with all regression results (Table 2, p.12; see Comment 3), we have decided to remove Figure 1 from the revised manuscript in order to avoid redundancies.

Comment 5: “The authors need to consider per capita stock of toilet paper, instead of stock, as this considers the natural demand for toilet paper given the household size. In the manuscript, it is not clear whether the authors consider a per capita measure or not.”

Response: We agree with the reviewer that it is important to account for variation in the demand for toilet paper due to household size. However, rather than dividing the amount of stocked toilet rolls by household size, our original analyses featured household size as a covariate. By doing so, any effects estimated for the psychological variables reflected incremental effects above and beyond variation in household size (and all the other predicters included in the baseline model). In response to the reviewer’s comment, we checked what would happen when, alternatively, a ratio variable (stocked toilet rolls divided by household size) was predicted in a linear regression (without considering household size as a covariate). Although the general pattern of results remained largely unchanged, there were also minor discrepancies when using this approach. However, while appreciating the reviewer’s concern, we chose to stick to the results obtained with our original approach (which also controls for variation due to household size) because predicting ratio variables in linear regression is not recommended for statistical reasons (Lien, Hu, & Liu, 2017). In particular, dividing two random variables reflects a non-linear transformation that creates a variable that potentially violates many assumptions of ordinary least squares regression. While there exist solutions to deal with these non-trivial statistical matters, we opted for our original approach of controlling for variation in toilet paper demand by treating household size as a covariate as this allowed to stick to one family of regression models for all dependent variables. Should you be still be concerned about the validity of this approach, we are also happy to explore results using a semiparametric model. 

Comment 6: “In 263-264 lines, do you mean participants who are more open to experiences stocked less toilet rolls than those who present lower level of openness? If so, rewrite these lines to make this point clearer.”

Response: We revised the Results section and focused on the main findings reoccurring across the different indicators of toilet paper consumption. Thus, this sentence was deleted from the manuscript. However, we report this particular finding in Table 2 (p. 12). 

Comment 7: “Can the authors provide an official reference for the content in line 41?”

Response: We added a source from the official German statistics office (Statistisches Bundesamt) which provides official numbers on the increase of toilet paper consumption in Germany based on market statistics (p. 2). 

Comment 8: “Despite the little variation of variables explained in lines 143-148, I recommend the authors to explain a bit more on which values/categories the data was concentrated the most.”

Response: We agree that additional information on these variables is valuable. In the revised manuscript, we now show the descriptive statistics for all variables by place of residence (p. 6; see Comment 1). Thus, this table includes the information requested by the reviewer. 

Comment 9: “Authors are finding associations, not impact or effects. The reviewer strongly recommends adjusting the language reflecting this (for instance, see lines 195 and 198).”

Response: We agree with the reviewer and have revised the language in the Results section to be more cautious in terms of causality. For example, critical phrases now read as follows:

• “The models for the HEXACO dimensions revealed a link between Emotionality and the perceived threat of Covid-19 [formerly: an effect of … on …]...” (p. 11) 

• Turning to the psychological predictors, the perceived threat of Covid-19 was positively related to all three ToP variables (ps < .025). (p. 12)

Note, however, that we have not changed the wording of the section that addresses the indirect effect of Emotionality on toilet paper stockpiling via the perceived threat of Covid-19. The reason for this is that these analyses do in fact assume associations to be driven by a certain causal directionality. Specifically, the assumption is that relations are neither caused by unobserved confounding variables (which we think is reasonable given the included control variables) nor caused by effects of opposite directionality (which is also reasonable given that toilet paper shopping is unlikely to causally increase threat perceptions or dispositional emotionality). However, we added a sentence to the discussion section highlighting the need for experimental data to warrant causal inferences without making such assumptions:

“Also, experimental studies would be required in order to explicitly test the directionality implied in our investigation of indirect effects.” (p. 20) 

Comment 10: “Could the authors clarify whether those people who tend to stockpile as a result of the uncertainty of the consequences and causes of the disease can be considered selfish? If not, where would this explanation enter the reasons outlined in pg. 3.”

Response: We thank the reviewer for this important note, as we realized that our writing was lacking clarity in this section. In the revised manuscript, we included another sentence in which we highlight the distinction between toilet paper consumption due to dispositional differences in prosociality vs. due to threat perception.

“Consequently, stockpiling toilet paper during the Covid-19 pandemic should be observed primarily among those who feel particularly threatened by the virus. Although stockpiling as a result of perceived threat might be considered selfish by some, it is important to note that it would not necessarily reflect a dispositional lack of prosociality. Instead, even the most humble and moral individuals might stockpile toilet paper as long as they feel sufficiently threatened by the pandemic.” (p. 3)

Comment 11: “Could the authors offer a brief discussion of what sort of framing governments could use when delivering messages to deter stockpiling during crisis?”

Response: We appreciate this suggestion and added a note about an empirical study by Jones et al. (2010) indicating the importance of clear communication in reducing panic in a population (see p. 17). 

“For instance, research on communication strategies suggests that clear communication aiming to increase awareness of a disease and providing simple behavioral instructions reduces people’s threat perception [26].”

We feel that this reference gives a good link to further studies about public communication in times of crisis and proposes clear measures for governments. From our view, however, our own data do not allow for concrete implications for governments since we did not assess, for example, the participants’ perception of the communication strategy of the government in their country of residence, or the specific communication of governments. 

References:

Lien, D., Hu, Y., & Liu, L. (2017). A note on using ratio variables in regression analysis. Economics Letters, 150, 114-117.

---

## [Editor Report · Decision Letter 1]

22 May 2020

Influence of perceived threat of Covid-19 and HEXACO personality traits on toilet paper stockpiling

PONE-D-20-10675R1

Dear Dr. Toppe,

We are pleased to inform you that your manuscript has been judged scientifically suitable for publication and will be formally accepted for publication once it complies with all outstanding technical requirements.

With kind regards,

Valerio Capraro

Academic Editor

PLOS ONE
---

## [Editor Report · Acceptance letter]

26 May 2020

PONE-D-20-10675R1 

Influence of perceived threat of Covid-19 and HEXACO personality traits on toilet paper stockpiling 

Dear Dr. Toppe:

I am pleased to inform you that your manuscript has been deemed suitable for publication in PLOS ONE. Congratulations! Your manuscript is now with our production department. 

With kind regards,

on behalf of

Dr. Valerio Capraro 

Academic Editor

PLOS ONE